# Perceived accuracy and utilisation of DHIS2 data for health decision making and advocacy in Kenya: A Qualitative Study

Phoene Mesa Oware[1,2]*, Gregory Omondi[1], Celestine Adipo[1], Mohamed Adow[1], Conrad Wanyama[1,3], Dan Odallo[1], Nelly Bosire[1], Lawrence Okong'o[4], Michuki Maina[1,3], Jalemba Aluvaala[1,3,4,5], Peter Ngwatu[1], David Githanga[1], Doris Kinuthia[1], Irene Amadi[1], Grace Rukwaro[3], Linet Kerubo[6], Cynthia Amisi[1], Allan Govoga[6], Janette Karimi[6], Ali Kahtra[7], Andrew Mulwa[6], Patrick Amoth[6], Ambrose Agweyu[1,3,4,8], Fred Were[1,4,5]

1 Kenya Paediatric Research Consortium, Nairobi, Kenya, 2 School of Public Health, University of the Western Cape, Cape Town, South Africa, 3 KEMRI-Wellcome Trust Research Programme, Nairobi, Kenya, 4 Kenya Paediatric Association, Nairobi, Kenya, 5 University of Nairobi, Nairobi, Kenya, 6 Ministry of Health, Nairobi, Kenya, 7 Council of Governors, Nairobi, Kenya, 8 London School of Hygiene and Tropical Medicine, London, United Kingdom

☯ These authors contributed equally to this work.

* pmoware@gmail.com

## Abstract

Reliable health information systems (HIS) are critical for effective decision-making in the delivery of Primary Health Care and Reproductive, Maternal, Newborn, Child, Adolescent Health and Nutrition (PHC/RMNCAH+N) services. In Kenya, the District Health Information Software 2 (DHIS2) platform serves as the primary HIS for tracking health indicators. This qualitative study explored perceptions of DHIS2 data accuracy and use for decision-making among PHC/RMNCAH+N stakeholders across 15 counties in Kenya. 89 Key Informant Interviews were conducted with PHC/RMNCAH+N stakeholders, to explore experiences, barriers, and facilitators of DHIS2 data use. Thematic network analysis was employed to identify recurrent themes and generate insights into the utility of DHIS2-generated information. Sociotechnical challenges included limited technical capacity among health staff, inadequate analytical skills, and reliance on a small pool of Health Records Information Officers (HRIOs). However, positive practices emerged, such as the use of DHIS2 dashboards and user-friendly outputs, which were valued for supporting evidence-based decision-making and advocacy, particularly at higher levels of health management. In some counties, visual displays of data, including scorecards and performance trends, facilitated budget advocacy and community engagement. Contextual challenges and constraints, such as use of inconsistent data collection tools across counties post-devolution, human resource shortages, and limited integration of private sector data, contributed to incomplete reporting. These challenges underpinned perceived inaccuracy of DHIS2 data, arguably, hindering the complete reliance on DHIS2 data

**Data availability statement:** De-identified data (text transcripts) can be availed upon request made in writing to: The Executive Director, Keprecon, through info@keprecon.org. This is because we did not have a clause to the participants to seek their consent for secondary data analysis. Therefore, the organisation will take the responsibility to protect the study participants.

**Funding:** PO, GO, CA, MA, CW, DA, NB, LW, MM, JA, PN, DG, DK, IA, GR, LK, CA, AG, JK, AK, AM, PA, AM and FW were supported by funds from the Gates Foundation, [# 033964), awarded to the Kenya Paediatric Research Consortium, (KEPRECON). PO received additional funding from South African Research Chairs Initiative of the Department of Science and Technology and National Research Foundation of South Africa (Grant No. 82769) for her post-doctoral research career development. The funders had no role in the design and conduct of the study; collection, management, analysis, and interpretation of the data; preparation, review, or approval of the manuscript; and decision to submit the manuscript for publication.

**Competing interests:** The authors have declared that no competing interests exist.

for planning and decision making. The study highlights the need for targeted investments to improve DHIS2 data accuracy and use through stronger stakeholder coordination, enhanced data synthesis skills, and fostering a culture of data ownership among a wide range of stakeholders in health, including political actors.. Addressing these gaps will contribute to improvement in DHIS2 data quality, enhanced ownership and reliance on DHIS2 data by PHC/RMNCAH+N stakeholders for decision making in Kenya.

## Introduction

Health Information Systems (HIS) are essential components of a health system, integrating data collection, processing, reporting, and use of health information to enhance efficiency and effectiveness of services [1]. A well-functioning HIS enables decision makers at all levels to identify challenges, engage in evidence-based decision-making, and optimize the allocation of limited resources [2]. Strengthening HIS is thus essential for tracking progress towards health goals, including Sustainable Development Goal (SDG) 3 – good health and wellbeing [3,4]. Concerns however remain over the capacity of HIS in many low- and middle-income countries (LMICs) to support decision-making. In these contexts, HIS often face challenges due to fragmented information landscapes, inadequate infrastructure, and limited resources [2,3].

The District Health Information Software 2 (DHIS2) is an open-source platform widely adopted for managing health data by at least 80 LMICs, at national and sub-national levels [5]. It can be customized to meet specific country/organizational needs; it is supported by basic hardware; and it offers easy analysis and real-time data visualization [6]. Kenya adopted and rolled out the DHIS2 system software nationwide in 2011 [7]. This coincided with a transition to a devolved system of governance, including health governance [7]. The previous structure, with 8 administrative units (provinces), was replaced with 47 administrative units (counties) [8]. In the devolved system, county governments are responsible for overseeing health services, including public health facilities that are organized into 4 levels: community services (level 1), dispensaries and clinics (level 2), health centres, maternity and nursing homes (level 3), and county referral services (level 4) [9]. The national government retained critical functions such as policy formulation, quality and standards control, and the management of national referral institutions (level 5) [9]. The devolution of health governance created a demand for data by county health managers who need to utilize DHIS2 generated information for decision-making including planning, budgeting, and monitoring programs, as well as among local elected leaders who rely on data to support their political manifestos and re-election bids [7].

While the technical and operational elements of DHIS2 are relatively well understood [6,10], it is not clear to what extent information generated via the platform is being used for decision making by different stakeholders in LMICS, and in the Kenyan context [11]. This study therefore investigates perceived accuracy of DHIS2 data and its utilisation for health decision making and advocacy in Kenya.

## DHIS2 data use for decision making and advocacy in sub-Saharan Africa

Evidence from various sub-national contexts in sub-Saharan Africa (SSA) shows the use of DHIS2 generated information for program review and planning, decision making and advocacy [11]. DHIS2 generated information is used to monitor indicators and evaluate district performance against national targets as seen in Tanzania and Cameroon [12,13]. In Sierra Leone [14], Ghana [15], and Tanzania [13], tracking of indicators performance and DHIS2-generated data charts and league tables has facilitated targeted actions and resource allocation. An example of DHIS2 information use in advocacy is seen in Sierra Leone, where practices such as ranking of sub-districts' performance on indicators have been found to foster dialogue and a sense of competition, subsequently encouraging local leaders to mobilize resources and enact policies to address low uptake of services in their constituencies [14].

Decision making based on DHIS2 generated information primarily occurs at the top levels - district and national - while at the facility level, data is predominantly collected rather than utilized [12]. In Tanzania, facility meeting reports in two districts revealed that at least 63% of facilities did not include data as an agenda item for discussion in their monthly review meetings [13]. In Ghana, [15] only 41.7% of district health management teams in the study region made decisions or shifted resources and strategies based on DHIS2 data comparisons, compared to only 19% at the community facility level.

Underutilisation of DHIS2 analytical tools poses a significant barrier to data use for decision making. Quantitative evidence suggests limited capacity to utilise DHIS2 analytical tools. For instance, in Ethiopia, about half of department heads in 83 facilities reported inability to calculate achievements based on targets [16]. In Kenya and Tanzania, findings from sub-national studies show that the proportion of facilities that could effectively use the DHIS2 analysis functions was 68% and 10% respectively [13,17]. In Ghana [15], a noticeable decline in capacity to analyse DHIS2 data was observed from district to community levels.

Across SSA, the accuracy of DHIS2 data varies significantly, with some countries and programs having more complete and accurate data than others [18–21]. DHIS2 data quality challenges relate to system design barriers such as inability of the software to accommodate certain health data [22,23]. However, the majority of DHIS2 data quality concerns stem from issues that occur during data capturing such as use of incomplete registers to collect data and inconsistencies in the calculation of indicators [21,22]. These challenges have been linked to organizational processes including weak implementation of standard procedures for reporting, use of incomplete reporting tools and inadequate supervisory support [7,12,16,18,24]. The (in)accuracy of DHIS2 data identified in many LMICs raises concerns over the reliability of DHIS2 data and willingness of decision makers to use it for decision making purposes.

Much of the existing literature on DHIS2 data use for decision making highlights experiences and perspectives of frontline health care providers such as health facility workers and data entry personnel. Few studies explore the perspectives and experiences of senior health managers. Moreover, other influential actors and decision makers in health, such as health champions who need data for advocacy and political leaders are often overlooked. This study fills these gaps by exploring how County Health Management Team (CHMT) members and other key stakeholders in health engage with DHIS2 data for decision making and advocacy across multiple counties in Kenya.

## A socio-technical approach to HIS data use

A social technical perspective on the use of technology, e.g., DHIS2 software in organizations, e.g., health management institutions posits that outcomes, such as evidence-based decision making, result from interdependent relationships between social/behavioural and technical sub-systems within an organization [25]. The social/behavioural sub-system (comprising elements such as members of an organization and relationships among them, organizational values, culture, goals) are recursively and reciprocally related to the technical sub-system (e.g., technologies such as DHIS2 software, techniques and skills required to operate them) [25,26]. As such, optimizing one sub-system alone, social/behavioural, or technical, without consideration for the other, results in the sub-optimization of the other or the socio-technical whole [27].

Optimal outcomes are realized when a 'goodness of fit' is achieved between social and technical elements within an organization [28]. The social and technical subsystems are however engulfed and are influenced by the environmental subsystem which is the political, economic and social context within which the social and technical sub systems operate [26].

We draw on this socio-technical perspective to explore how social, technical and environmental aspects shape DHIS2 data use for decision making and advocacy in Primary Health Care (PHC) services with a focus on reproductive maternal child and adolescent health and nutrition (PHC/RMNCAH+N) in Kenya.

## Materials and methods

### Ethics statement

Ethical approval to conduct this study was obtained from The African Medical and Research Foundation (AMREF) Ethical and Scientific Review Committee (P1258-2022). Permission to conduct the study was also sought from the National Commission for Science, Technology, and Innovation (NACOSTI) (permit number NACOSTI P/17/1976) in Kenya. Written informed consent was sought from all participants before the interviews. Consent was reaffirmed at various points during the interviews. Several measures were taken to ensure the safety and comfort of study participants including informing participants of their right to withdraw from the study at any given point. Participants' confidentiality was further guaranteed through anonymisation of their identities in all research outputs.

This qualitative study was conducted as part of the Championing Evidence Based Advocacy (CEBA) project. The project aimed to strengthen and harness the capacity of health professional associations and local champions to advocate for increased use of data in decision making, planning & resource allocation for PHC focused on RMNCAH +N in Kenya.

### Study site and participants

The study was conducted in 15 counties in Kenya, namely: Busia, Vihiga, Migori, Kisii, Nairobi, Nakuru, West Pokot, Meru, Tharaka Nithi, Kirinyaga, Nyandarua, Mombasa, Taita Taveta, Mandera, and Wajir. County selection criteria, aimed to ensure regional representation and included counties with both high and low performance on tracer PHC/RMNCAH+N indicators across four quarters in 2022 as captured in DHIS2 (S1 Fig).

The majority of the population in the included counties reside in rural areas [29]. Key reproductive, maternal, child and adolescent health indicators show substantial variation across the counties; the unmet need for family planning ranged from 6.7% to 30%; women aged 15–49 with four or more ANC visits for their most recent live birth ranged from 35% to 75%; full vaccination coverage (basic antigens) among children age 12–23 months ranged from 49% to 96%, and prevalence of stunting among children under 5 ranged from 10% to 34% [30]. Detailed information on County profiles is presented on Table 1.

Representation of counties with diverse characteristics enabled understanding of the experiences and dynamics influencing DHIS2 data use from different perspectives.

According to Kenya's data quality assurance protocol 2014 [31] and analyses of DHIS2 Health Information flow [32], data reporting and quality control is envisioned as follows: Community Health Workers (CHWs), Community Health Extension Workers (CHEWs) and facility workers, e.g., Health Care Providers (HCPs) collect data at the community and facility levels respectively. Health Records Information Officers (HRIOs) working at the facility level or at the sub-county level in cases where hospital do not have HRIOs aggregate data into summary tables, and into DHIS2 on a monthly basis. Stakeholders, at all levels, that is, CHWs and CHEW, facility level health workers, facility managers, and CHMTs are expected to provide feedback on data quality to HRIOs to ensure completeness and timeliness in reporting.

In principle, all collected data must be analysed, disseminated, and used for feedback and decision- making at all levels of the health system [32]. CHMTs, county health departments and facility managers whose role is to coordinate health affairs in the county and the facility levels respectively can use data to monitor trends such as of disease outbreaks,

**Table 1. Population and key reproductive, maternal, child and adolescent health indicators of included counties.**

| | | Population | | | | | Percentage of currently married women age 15–49 with unmet need for family planning, | Women age 15–49 with four or more ANC visits for the most recent live birth | Percentage of children age 12–23 months who were fully vaccinated (basic antigens) at any time before the survey | Percentage of children under age 5 who are stunted |
|---|---|---|---|---|---|---|---|---|---|---|
| | | | M | F | Urban | Rural | | | | |
| Busia | Western Kenya | 426252 | 48% | 52% | 13% | 87% | 18.6 | 70.7 | 82 | 15 |
| Kirinyaga | Central Kenya | 302011 | 49% | 51% | 24% | 76% | 6.7 | 67.6 | 87 | 11 |
| Kisii | South Western | 605784 | 48% | 52% | 12% | 88% | 14.9 | 62.7 | 90 | 16 |
| Mandera | North Eastern | 434976 | 50% | 50% | 31% | 69% | 17.3 | 40.4 | 29 | 21 |
| Meru | Eastern | 767698 | 50% | 50% | 8% | 92% | 7.8 | 45 | 82 | 25 |
| Migori | South Western | 536187 | 48% | 52% | 15% | 85% | 20.1 | 58.5 | 86 | 15 |
| Mombasa | South-eastern | 610257 | 51% | 49% | 100% | *- | 19.1 | 65.3 | 93 | 14 |
| Nairobi | Southcentral Kenya | 2192452 | 50% | 50% | 89% | 11% | 12.5 | 35 | 77 | 11 |
| Nakuru | Rift Valley | 1077272 | 50% | 50% | 48% | 52% | 8.3 | 73.4 | 92 | 19 |
| Nyan-darua | Central | 315022 | 49% | 51% | 10% | 90% | 8.6 | 60.6 | 95 | 18 |
| Taita Taveta | Coastal region | 173337 | 51% | 49% | 28% | 72% | 12.8 | 64.9 | 85 | 19 |
| Tharaka Nithi | Central | 193764 | 49% | 51% | 8% | 92% | 9.7 | 63.2 | 95 | 21 |
| Vihiga | Western | 283678 | 48% | 52% | 10% | 90% | 18.3 | 74.9 | 96 | 17 |
| Wajir | North Eastern | 415374 | 53% | 47% | 23% | 77% | 12.7 | 44.9 | 49 | 12 |
| West Pokot | North Rift | 307013 | 49% | 51% | 5% | 95% | 30.3 | 35 | 49 | 34 |

*Rural population for Mombasa not captured in census data

identify gaps in service delivery, plan interventions and allocate resources [31]. Legislators have a responsibility to scrutinize health policies and approve budgets for health services-informed decision making [33]. Non-state actors can support the institutionalization of HIS and implementation of quality standards [31]. In addition to utilizing data for decision making, CHMTs have an explicit role of analysing data [31].

We purposively sampled and interviewed 89 key informants that are involved in health decision making/advocacy roles. They comprised members of the county health executive, that is, CHMT members and members of health departments, county legislature (Members of County Assembly-MCAs), and non-state actors, who included representatives of health professional associations, representatives from non-governmental organisations (NGOs), Community Based Organizations (CBOs) and human/health rights champions. Study participants were recruited by the CEBA project county focal persons (CFPs) who were mainly paediatricians, and supported by research assistants (junior clinician and non-clinician health workers) and community health workers.

## Data collection

All interviews were conducted by CFPs assisted by research assistants, both of whom received training in qualitative research methods, facilitated by experienced qualitative researchers at KEPRECON in March 2023. The training covered qualitative methodology, research ethics, participant recruitment, interviewing skills, reflexivity, fieldwork logistics, and transcription.

Data were collected between March, 20th and May 16th 2023. To prompt discussion on perceptions on DHIS2 data quality and use, participants were presented with graphs showing their county's performance on PHC/RMNCAH+N indicators over four quarters of 2022. They were asked to reflect on the accuracy of the trends. This was followed by a discussion on how they used DHIS2 data to support decision making and advocacy for resources to implement PHC/RMNCAH+N policies. Both CFPs and research assistants wrote reflective notes using note taking templates during and after each interview to summarise key themes and context details. Interviews lasted between 23 and 97 minutes. Weekly debriefing sessions were held with the research project leads to provide supervision, reflect on interviewing approaches, explore emerging themes, and adjust focus for subsequent interviews. Interviews were conducted in English and Kiswahili. Interviews were audio-recorded, transcribed, and translated verbatim into English by research assistants. 4 Quality control officers with more experience in qualitative research methods reviewed all transcripts to ensure accuracy in transcription and translation.

## Data analysis

Thematic network analysis [34] was used for data analysis, aided by Atlas.ti (version 23.2.3). Transcripts were coded both inductively, based on salient themes emerging from the data and deductively, guided by the study objectives. An initial codebook was developed based on the research objectives and preliminary transcript review. Subsequently, inductive coding was applied to allow for the emergence of new themes. Issues discussed by participants with respect to the study objectives within each code or set of related codes were then identified and grouped into basic themes. Labels considered as basic themes reflected simple, lower order patterns that were visible in the data. Related basic themes were then clustered under broader organising themes. Organising themes depicted higher order patterns in the data. Labels assigned as organising themes illustrated ways that several basic themes pointed to specific overarching issues/factors, e.g., that were linked to broader systems and processes. Organizing themes were subsequently clustered into global themes, each encapsulating the main idea within a large portion of data in relation to the research objectives. Each thematic network comprising basic themes, organizing themes and a global theme was read and explored in the context of the data, to ensure that it groundedness in participants accounts.

Several strategies were employed to enhance trustworthiness of the findings, including validation of the initial codebook by the technical research team (CW, DO, AA, FW) before its application to the remaining transcripts. Transcripts were independently coded by three researchers (PO, CA, MA) to enhance inter-coder reliability. Regular debriefing sessions were held with project technical leads to discuss emerging themes, share individual viewpoints and positionalities, and mitigate potential biases in the interpretation of findings. Findings were discussed during a validation workshop that was held in October 2023 and was attended by CFPs, CHMTs from participating counties, representatives of health professional associations, champions, among other key stakeholders in health - some of whom were study participants. During this session, respective county findings were shared for sense-checking, affirmation or seeking for more relevant contextual interpretation to address any perceived researchers' biased viewpoints.

## Reflexivity statement

Several factors should be considered when interpreting this study's findings. Firstly, as insiders within their study contexts, CFPs were able to establish rapport with CHMT members, which facilitated the gathering of rich insights. Their insider

knowledge and roles as frontline implementers of PHC/RMNCAH+N policies also enabled nuanced discussions with study participants. However, in a few cases, CFPs interviewing CHMT members (often their administrative superiors) about awareness of PHC/RMNCAH+N indicator trends and data use was perceived as interrogation or insubordination. It is possible that this may have shaped some participants' willingness to openly engage in the interview or to present their actions on data use for decision making more favourably if they perceived the interview as scrutiny. Additionally, participants were presented with trends of their counties' performance on various PHC/RMNCAH+N indicators as a conversation starter. We acknowledge that in so doing, researchers may have introduced a priming effect. Foregrounding specific trends may have influenced the direction of the discussions, for example, directing attention to issues that they many not otherwise have raised while diverting attention from others. This may also have triggered defensive reactions if trends presented contradicted their views on data accuracy. This was an intentional strategy, which aligned with the study's aim. Interviewers encouraged participants to reflect more broadly about issues that shape DHIS2 accuracy and their engagement with data use beyond the trends that were presented.

## Results

### Characteristics of study participants

Participants included: 50 CHMT members from all 15 participating counties, 18 members of county legislatures specifically, members of county assembly, and 16 non-state actors. Participants also included three facility managers and two facility wokers. Table 2 below presents detailed participants' characteristics such as sex, educational attainment.

A minimum of three CHMT members were interviewed in all counties. However, for the remaining participant categories, namely, county legislators, facility managers and non-state actors, representation varied. This was based on participants' willingness and availability as well as judgement by CFPs on who was best suited to address the study objectives in their county context.

Even though we did not seek balanced representation of participants by sex, some patterns stood out. In one predominantly urban county, all key informants were female, and in another, all but one participant was male. In one predominantly rural county, all key informants were male, and in another, all but one was female. 40% of interviewed participants were female and 60% were male.

### Thematic findings

Findings are presented under two global themes (Table 3). The first global theme, use of DHIS2 data and information by CHMT members for decision making and advocacy is discussed under three organizing themes: *review of DHIS2 data*, *synthesis of DHIS2 data* and *capacity to synthesise DHIS2 data.* These themes explore socio technical influences on the use of DHIS2 generated information. Findings on the second global theme: perceptions of DHIS2 data quality are discussed under three organizing themes that explore contextual dynamics that influence perceptions of DHIS2 data quality among CHMT members, these are; (a) *Devolution and use of outdated or improvised registers* (b) *(Non) Representativeness of DHIS2 data* and (c) *Human resource constraints to monitor indicators*.

To maintain anonymity and confidentiality, the names of counties have been de-identified and assigned numerical codes, and details such as participants' age, sex, and years of experience omitted.

### Use of DHIS2 data and information by CHMT members for decision making and advocacy

**Review of DHIS2 data.** Across all counties, participants reported routinely reviewing DHIS2 generated information at the county and sub-county levels to inform decision making. They identified several gaps that undermined this practice, such as lengthy reporting intervals, lack of follow-up on actions, limited engagement with data at the facility level, and restricted access to the DHIS2 platform by stakeholders at operational levels.

**Table 2. Key informants interviewed and their functions in relation to DHIS data use.**

| Stakeholder category | Key informants interviewed | Sex | Highest qualification | Total |
|---|---|---|---|---|
| Development Partners and Implementing Partners | **NGOs, CBO and FBO representatives from 10 counties**<br>1. Human Rights Advocate<br>2. Human Rights Defender<br>3. Project Coordinator (CBO)<br>4. Programme Health Coordinator- NGO<br>5. Community Based project Coordinator-NGO<br>6. Project Officer- Women with Disabilities group<br>7. Member of Civil Society Organization<br>8. Health champion<br>9. Representative-Health CSO coalition<br>10. Faith Based Organisation Representative<br>11. NGO rep-Nutrition Support Officer<br>12. CSO representative - Manager Development<br>**Members of Health Professional associations from 4 counties**<br>13. NNAK representative<br>14. National Nurses Association of Kenya (NNAK) representative<br>15. Kenya Clinical Officers' Association (KCOA) representative (treasurer)<br>16. Health Professional Association (OBS/GYN) | F – 62%<br>M – 38% | University degree – 50%<br>College diploma – 43.75%<br>Primary – 6.25% | 16 |
| County legislature - MCAs | **From 13 counties**<br>1. 4 Members of County Assembly<br>2. 6 MCAs and also Chairpersons of County Health Committee<br>3. 2 MCAs and also chairs of finance committee<br>4. County Budget officer<br>5. Member of County Assembly (party minority leader)<br>6. Vice chairperson health committee<br>7. MCA and member health committee<br>8. MCA- Budget Committee<br>9. MCA- Chair Budget Appropriation Committee | F– 17%<br>M– 83% | University degree – 78%<br>College diploma – 22% | 18 |

*(Continued)*

| Stakeholder category | Key informants interviewed | Sex | Highest qualification | Total |
|---|---|---|---|---|
| County Executive: County Health Management Team Members and Representatives from County health departments | **From all 15 participating counties**<br>1. 9 County Reproductive Health Coordinators<br>2. Sub County Reproductive Health Coordinator<br>3. Head of reproductive health<br>4. County Reproductive health officer<br>5. Chief Health Officer-Health<br>6. Director of Curative and Preventive Services<br>7. County Director of Universal Health Care<br>8. 4 County Directors of Health<br>9. County Director of Health Services<br>10. Deputy Director of Health<br>11. County Director of PHC Services<br>12. Director of Programs<br>13. County Nutrition and Dietetics Coordinator<br>14. 2 County Nutrition Coordinators<br>15. Chief Officer Medical Services (also acting chief officer of nutrition wellness in school feeding programme)<br>16. County Nutritionist<br>17. County Nutrition and Dietetics Officer<br>18. County Child Health Coordinator<br>19. Chief Officer for Public Health<br>20. 3 County Directors of Public Health Services<br>21. Deputy Director Public health<br>22. Chief Officer public health<br>23. County Chief Officer of Health<br>24. Director Health Department<br>25. County Health Promotion Officer<br>26. County Health Promotion Coordinator<br>27. County director preventive and promotive health<br>28. County RMNCAH coordinator<br>29. Ministry of Health-Sub County Head<br>30. Primary Health Coordinator<br>31. County Nursing Officer<br>32. CECM Health and Medical Services<br>33. County Health Promotion Officer<br>34. Deputy Director Health Administration<br>35. 2 County Community Health Services coordinators | F– 44%<br>M– 56% | Post graduate – 10% University degree- 74% College diploma- 16% | 50 |
| Facility Management Teams | **From 3 counties**<br>1. Hospital Manager<br>2. CEO-County Teaching and Referral Hospital<br>3. Medical Superintendent- County Referral Hospital | M 100% | All with university degree | 3 |
| Facility health workers | **From 2 counties**<br>1. Nurse- Ripples international- NGO<br>2. Nurse Officer in charge- FBO | F – 50%<br>M– 50% | All with university degrees | 2 |
| | | | | **N = 89** |

CHMT members in all counties narrated that review of DHIS2 data informed resource allocation, design, and implementation of targeted interventions, e.g., staff continuous medical education or community interventions to address poor indicators. In one example, elaborated in the quotation below, review of DHIS2 information revealed disparities between population and facility indicators. This suggested that residents sought healthcare services outside the county. Subsequently, the County Integrated Development Plan (CIDP) prioritised establishing a level four hospital in each sub-county.

**Table 3. Thematic networks.**

| Global themes | Organising themes | Basic themes |
|---|---|---|
| 1. Use of DHIS2 data and information by CHMT members for decision making and advocacy | a. Review of DHIS2 data | • Long reporting intervals<br>• Lack of follow-up actions<br>• Minimal engagement with data at facility-level<br>• Restricted DHIS2 platform access |
| | b. Synthesis of DHIS2 data | Performance trends illustrated using scorecards, dashboards |
| | c. Capacity to synthesise DHIS2 data | • Over reliance on HRIOS for analysis<br>• Building health teams' capacity for data analysis<br>• Establishing health research units |
| 2. Perceptions of DHIS2 data accuracy and factors influencing perceptions | a. Perceived accuracy of DHIS2 data | • Doubts regarding data accuracy<br>• Distortions of data<br>• Data differs from 'reality on the ground' |
| | b. Devolution and use of outdated or improvised registers | • Inaccurate data capturing<br>• Shortage of reporting tools<br>• Improvisation of data capturing tools<br>• Use of outdated registers that miss some indicators |
| | c. (Non) Representativeness of DHIS2 data | • Exclusion of services by private pharmacies<br>• Private providers' reporting rights on DHIS2<br>• DHIS2 focus on facility data, neglecting community-level incidents<br>• Difficulty in tracking migration of service users<br>• Strengthening routine data quality assessments and audits |
| | d. Human resource constraints to monitor indicators | • HRIOs shortages<br>• Task shifting by health teams<br>• Backlog of unprocessed data<br>• Limited HRIO training on aspects of data management |

*So, when […] we're in the budgeting and planning seasons, our plans are usually based on DHIS based data. And in fact, […] even the […], the current CIDP, the issue of … ah, providing level four services at the sub-county level, has actually been necessitated by this observation in our data. Where we are, […] picking that a lot of our clients are seeking services outside the county. Because our population-based data, indicates that we are doing relatively well. But when we look at service delivery data, it tells you we are not the ones who are offering those services, somebody else is. So … and we looked at the facilities, the […] sub-counties which are giving us poor indicators, and we noted there are those sub-counties that don't have hospitals. And it has become our -- our number one priority now, to establish […] level four hospitals in each of the sub-counties. […] a serious policy, planning and budgetary […] you know decision. (Deputy Director of Health, County 10)*

Experiences of legislators using data for decision making were however few. Specifically, they were shared by MCAs in two counties. One, described using data to demand a specific intervention by the department of health as follows:

*Well, majorly we sit down to look and […] analyze, so that we know where to invest and what laws to come up with, yes. […] On […] adolescent health issues […] that's why we even--we actually requested the Department of the Public Health to come up with the condom dispensers across the county at the streets, […] where people can be able to access them. Okay and they make sure that they be refilled. So, we had data in terms of the rate of infections. […] (Member of County Assembly and Chair health committee, County 9)*

Echoing the same experience, another MCA from county 2 indicated that having data helped them (legislators) to argue for investments in specific health programmes in county assembly '*from a point of knowledge*' but stated that sometimes, they did so '*without having the data*'.

CHMT members identified the need to follow up on actions agreed upon during data review meetings in order to address poor indicators. In some counties, CHMT members also discussed the need for timely data capturing and real-time uploading (as opposed to monthly as seen in one county-13), to support timely evidence-based actions. For example, early detection of outbreaks and formulation of timely responses.

> *Health needs to get to a point that […] our health information system is real time. That we'll be able to […] improve the responses. For instance, […], if we have an outbreak somewhere, by twenty-four hours somebody looking at the data would actually have seen the variation. But you know we up-upload at the end of the month. (Deputy Director of Health Administration County 13)*

Participants in several counties perceived limited ownership and use of DHIS2 information at the facility level. CHMT members, from Counties two, six, thirteen and fifteen noted that dialogue about indicators needs to extend beyond senior health management teams, to include facility and community-level stakeholders. The County Director of primary health care services from county two stated:

> *so, at the end of the month, once we've handed in their [facilities'] reports, I think it's important to […] throw back the same data to the facility so that they can also see how they are doing. And so, that way, even they also consume their own data. I think there is that mechanical…uh… point of view, where you're processing the data for […] the senior most person, but at the point where you are processing this data (background chatter) you are not also consuming the data. So, I wish we could trickle down, we– we — […], share it downwards. The information comes up, but it needs to go back down so that the facilities […] really are able to improve on the places where they're not working well.*

An important step that was highlighted by participants to address limited engagement with data at the facility level was granting access to the DHIS2 platform to all relevant stakeholders including HCPs. For example, the sub county reproductive health coordinator from County 9 noted that currently, only the HRIOs had database access rights. She explained that expanding access to the DHIS2 software platform could enable HCPs to review data at their convenience, stay informed about data trends, and proactively address gaps.

> *because we don't have rights to the DHIS, It's only the HRIO [Health Records Information Officers] who has the rights. So, for me to analyse that data, I have to look for the HRIO. But if I had that data, and on and off I can sneak peak … they can even block it, like we cannot alter any data right on our end but it's a read only, so that at a click of a button I can tell what is happening in the sub-county and then maybe at that point I can do something. But for me to do, like if you look at my dropout chart is not filled for this month. I have to wait for my HRIO to give me data so that I can be able to, so this one delays…*

The issue of access to health data was raised by one legislator who noted that access to data could be helpful for decision making by legislators.

> *At times we talk without having the, the data. [….] it's not easy at times to access the data. So, having a data and an updated one, […] will help a lot. Even in policy making, you know where we have gaps, you know where we are doing well, you know where and when to improve. (Member of County Assembly, County 2)*

He described depending on health departments to furnish them (legislators) with data, stating that '*as an-a county assembly, we rely so much--every, [...] every information comes from the department itself to furnish the county assembly*' and emphasized the need to strengthen this collaboration.

**Synthesis of DHIS2 data.** Participants narrated that synthesis of DHIS2 data into user-friendly outputs facilitated engagement with DHIS2 data by a wide range of stakeholders and supported advocacy for increased investment in PHC/RMNCAH+N programs.

In County 7, the director of public health described that illustration of performance trends across different indicators made it easier for CHMT members to defend health budgets and secure additional resources from the County government to implement PHC/RMNCAH+N programs. She narrated:

*You see when data is analysed like this [i.e., depicting trends as was done in documents presented to participants during the interview], it is more palatable. […] If you show anyone, they can see a trend […]. It speaks a lot too. […] I think that's what we are trying to do now. We are doing a lot of data analytics. […] And […] we try to do that to also convince. Like for this year, when we went for budget […] defence, […] we managed to get an extra five hundred million. Hence, if you defend, you know your budget, using the data that you've had. […] That our indicators have really gone down, because of this and this, so you are given an additional funding.*

In counties 2 and 15, CHMT members observed that indicators, such as presenting data on teenage pregnancies by wards (the lowest electoral unit), could influence MCAs to prioritise addressing poor indicators in their respective wards. In this regard, the Deputy Director of Public Health from County 6 described their use of a scorecard system, with colour-coded indicators representing different performance levels for each ward.

*we have been able to develop what we call a scorecard […] On quarterly basis, the scorecard brings out all the RMNCAH indicators -- if it is pregnancies, it is maternal, all the RMNCAH indicators. And we have been able to develop this per ward such that if you are the MCA of ward A, and we develop that scorecard, we come we invite you and members of that ward, critical stakeholders on that ward for a meeting. We might let you know your performance. And in the scorecard, we have even those […] regions which are, […] red, red, red. If your ward is red, in most of the indicators, you know you have a problem. So that has really helped us to do advocacy, and also make these critical people like the MCAs really understand that they must do something to change their red to a yellow or green.*

As is elaborated in the quotation above, scorecards served as user-friendly informative tools that enabled quick assessment of each ward's performance by MCAs. This fostered dialogue between the county health executives and the Members of the county legislature to address poor PHC/RMNCAH+N indicators.

Participants from counties 2, 3, and 4 also recommended the implementation of colour-coded dashboards on the DHIS2 software across all health programs, for easier data visualisation of performance. They noted that dashboards, which employ colour-coding to represent performance levels on various PHC/RMNCAH+N indicators facilitate faster comprehension and enhance utilisation of data by health professionals, CHWs, health advocates, champions, and policymakers. The County Nutrition and Dietetics Coordinator from County 3 narrated:

*I think we should come up with dashboards for specific programs in KHIS [Kenya Health Information System]. The dashboard is just a… at a glance you will be able to see what is happening. Other than figures, […]. But if there is a dashboard, you know, green means this, red means this, you know, […] yellow means this, then at a glance then you will be able to see yes, my area is not doing well, this indicator is not doing well. So, we have some areas like HIV and TB and… RH [reproductive health] having dashboards, and it is very easy to work in that area. But if we have all other*

*programs in KHIS having their dashboard, and dashboards you know, linked properly with the data collected in KHIS, then we will be able to…to use it properly.*

The above narratives highlight that alignment between DHIS2 data outputs and end users' preferences, capabilities and interests facilitated engagement with data by a wide range of stakeholders and supported advocacy initiatives. Notably, experiences of producing and sharing user-friendly outputs were shared in a few counties and across limited health programs, highlighting a need for replication of these practices.

**Capacity to synthesise DHIS2 data.** Despite recognising the importance of synthesising DHIS2 data for decision-making and advocacy, CHMT members in different counties expressed concern over limited existing capacity within counties to do so.

Participants from counties 6 and 10 for example suggested that enhancing the capacity of CHMT members to access, analyse, and synthesise DHIS2 data can reduce reliance on HRIOs for data analysis and interpretation.

*First and foremost, we have very few people who know how to use DHIS. To be ascertain [precise], I think it is only HRIOs, and very few people in management, and what those in management can only do is just to look at what is entered. But we need to have several people, especially those in management who… to be taken through […] how they can analyse data, so that even if data is put into figures, how can you put them into tables and the rest and analyse and make some decision. That is one thing. Because you have only like HRIO, mostly, eh mostly the HRIO we can say are the only ones who can do several things and generate tables and graphs like these (papers shuffling) [referring to graphs presented by interviewer] and he can tell you, "This […] one you are doing badly." Others, we can only access. (Primary Health Coordinator, County 10)*

The County Reproductive Health Coordinator from County 1 added that establishing county health research departments can further enhance the capacity of county health departments to understand dynamics that underlie poor indicators.

*we need continuous research. In this county, we don't have a research department […]. And that's where we are not doing well. We are depending on external research, like the one you're doing, but as a county we don't have a research department. It's [if] only County 1 can have research departments, we should be able to identify our own gaps, be it at the community level, be it at the facility level, be at the managerial level […] […] So those are the things that we want. I know that research is expensive. But that's the […] best.*

These findings revealed a dependency on a limited pool of experts, i.e., HRIOs for DHIS2 data analysis and interpretation, highlighting the need to strengthen capabilities of other members of health teams to access, synthesise, and interpret DHIS2 data.

### Perceptions of DHIS2 data accuracy and factors influencing perceptions

**Perceived accuracy of DHIS2 data.** Participants were asked to reflect on whether the PHC/RMNCAH+N trends that were shown to them were accurate, i.e., if they aligned with their perception (based on alternative data (if any) or intuitive sense of how their county was performing. CHMT members and non-state actors expressed mixed views regarding the accuracy of the trends presented. Many had some doubts.

*I think, to a certain degree there is a level of accuracy but not 100%. Because we have also compared the—the--eeh raw data in the registers and the summary data, many times we find the raw data in the registers have some gap with*

*the summary data. These gaps, there may not be extremely [huge] gaps. So, that's why I am saying the present [data presented] they are somehow… they're not 100% accurate, but they it's a reflection of the real thing. (Representative of a health professional association, County 8)*

*HPV vaccine. Mmm… (smacks lips) why do I feel this might not be right. Because actually what we know is we got a trophy over this HPV. So, eeh, probably the issue of data entry, but I know HPV we are doing well. (Reproductive Health Coordinator, County 10)*

Three legislators reflected on DHIS2 data accuracy. Of these, two felt it was accurate and highlighted indicators which they found to be alarming. Echoing the sentiments expressed by the reproductive health coordinator quoted above, one shared concern about the data being distorted, failing to reflect efforts on the ground.

*Because you do so much on the ground but then now when you go to get data […]. At times you get data, but it... you find it [is] a bit distorted. […] We have done so many programs uh with different or- organizations, but really, (background noise) they don't come back to us to give us feedbacks or, or what […] has happened. So, I think this is where we have a--a big gap. […] So, we don't know whether we are actually making any impacts with the programs we are running or not? (Member of County Assembly, County 2)*

Participants were not asked, and neither did they reflect on whether their perceptions of DHIS2 data accuracy shaped their willingness to use it for decision making and advocacy. These perceptions merely reflect some level of distrust in health data that is available in their context.

CHMT members explained that inaccuracy of DHIS2 data was due to contextual factors, namely, devolution of health services and consequently, use of outdated or improvised registers, (Non) Representativeness of DHIS2 data as existing data collection mechanisms failed to account for patient transfers and service provision by private providers and human resource constraints to monitor indicators. We delve into these in the sections below.

**Devolution and use of outdated or improvised registers.** CHMT members narrated that use of inappropriate reporting tools contributed to inaccurate data capturing. In two counties CHMT members described occasional shortages of reporting tools which led to improvisation of data capturing tools. In County 8, participants narrated that data gaps emerged due to the use of outdated registers that lacked provisions for capturing some indicators. They noted that the county had not yet developed its own data capturing registers following the devolution of health services, which had shifted this responsibility to the county level. Similarly, in County 7, participants noted that the use of outdated registers (which had since been resolved) had led to underreporting on the postnatal care indicator in previous quarters.

*there are some things which are pulling us back […] because of the registers, you find that we have the revised registers. But […] national [government] just provided a few since health is already devolved// […] County is supposed to be now making registers for us. But it's a challenge. […] So, you'll find that they have gone back [to] the old registers… where we don't have proper classification. We don't have some of the indicators in these old registers. (County Child Health Coordinator, County 8)*

Improvisation and the use of old registers not only led to missing data on some indicators, as elaborated in the above quotation, but can also contribute to variations in monitored indicators across counties and produce incomparable data at the national level.

**(Non) Representativeness of DHIS2 data.** Participants perceived gaps in DHIS2 data due to inadequacy of existing data collection mechanisms to track and patient transfers and services provided by private providers.

CHMT members in two counties indicated the actual uptake and coverage of family planning services could be higher than what was reported in the DHIS2 records, which did not account for services offered by private pharmacies, colloquially referred to as chemists.

*if, the National [government] can work a way out whereby even the private pharmacies and chemists can be given rights to report to the national reporting platform. […] And for them to have the rights to report that facility pharmacy or whatever you're calling it must first of all, have a KMFL (Kenya Master Health Facility) number. […] if they don't have the KMFL number they can't report but still there is also something else that can be done, they can be linked to a public facility. Two we tried that, and it was, and we saw that it can kind of work but sustainability was an issue. (County RMNCAH Coordinator, County 8)*

As described by the CHMT member from County 8 above, granting private providers of family planning commodities reporting rights to the DHIS2, or linking them with public health facilities to report their data is important to address under-reporting of data. However, as elaborated in the above narrative, ensuring the sustainability of such strategies at the county level is challenging, indicating a need for national-level endorsement and support to implement such strategies.

In two other counties, participants indicated that PHC/RMNCAH+N indicator trends presented to them overstated the counties' performance since they primarily captured health facility incidents but overlooked community incidents, e.g., of malnutrition, infant and maternal mortalities, and unskilled deliveries. The County Nutrition Coordinator from County 15 indicated that health facilities in the County were underutilised.

*The only challenge that we have with the DHIS data is the coverage. How many people come to the health facility? So, our [facility] coverage is less than 30%. So now you will realise sometimes when you have a low coverage then in the… the…. the indicators like for example how many children are malnourished will be below…way below what is actually on the ground. So, it is only the coverage.*

Participants further described migration of mothers to different locations or facilities, as seen in County 11, and their preference for accessing skilled delivery services in neighbouring counties, as seen in Counties 10 and 13. This, along with the challenge of tracing and documenting these movements, made it difficult to accurately capture data on child immunisation and skilled delivery services. The County Community Health Services Coordinator in County 13 highlighted data discrepancies that arose from capturing facility transfers by pregnant women as 'defaulters' in facility records, even though data from CHWs indicated no default rates.

*what happens is that, when you go to our facilities, you will realise that we have ANC defaulters. But when you come to us in the community, we rarely have those defaulters. What happens is that a mother may decide to change a health facility.*

The challenges related to the migration of mothers between facilities/locations highlight the complexities of tracking PHC/RMNCAH+N service utilisation and the need to improve data capturing methods to account for such movements. To this end, CHMT members emphasised the need to strengthen routine data quality assessments/audits through processes such as cross-referencing data sets. They noted that these assessments should involve all stakeholders, including CHWs to interrogate and evaluate the consistency between data recorded at the community and health facilities. Participants in County 13 pointed out that inclusion of CHWs in data audits can strengthen monitoring of PHC/RMNCAH+N indicators within their communities by providing an accurate account of whether mothers have defaulted on their appointments or transferred to other facilities. In County 10, the Deputy Director of Health noted that CHWs also conduct verbal autopsies

within communities which contribute to a deeper understanding of the circumstances surrounding some indicators such as child mortality.

*we should also intensify. […] Routine Data Quality Assessments, […] moving on the ground also to compare, between what we find in registers and what we find in summary, in summary forms, and what we find in, find in, in tally sheets and stuff like that to see whether… […] it improves on the quality of data.*

In County 9, participants highlighted an effective data capturing mechanism from the household level through CHWs to the facility level. This involved providing documentation tools to CHWs to record data at the community level. A Community Health Extension Worker (CHEW) would then compile this data into a report, which was subsequently entered into the DHIS2 system. Occasional shortages of reporting tools however led to inconsistencies between CHWs' and facility records.

*The tools are not adequate. […] Inadequacy of tools and therefore sometimes they [CHWs] will refer [patients] and then not capture. […] But the service was offered. […] But at the facility level we can only see an increase. […] But maybe in the community tool you may not be able to capture and relate. (County Community Health Services Coordinator, County 9)*

To mitigate the shortage of physical data capturing tools, CHMT members in Counties 5, 12, 13 and 3 recommended the adoption of electronic data capturing methods, along with the provision of necessary of necessary infrastructure, a practice that was already being implemented in County 6. The Deputy Director of Health in County 6 highlighted the successful implementation of an electronic medical records system across 12 sites. This system, which was utilised by over one thousand CHWs in the county, enabled updating of data by CHWs via mobile phones and ensured smooth integration of data collected at the community level into DHIS2.

*We have […] I think 12 sites, where we are doing […] electrical medical records (EMR) within the county […]. We also have for our community health volunteers, we have […] slightly over… above 1000. […] currently using the digital platform, currently using phones to report on matters. Whenever they go to their houses, they report everything digitally. And this digital platform is anchoring what we call each is a national platform […] And reports are just generated now from KHIS through… it doesn't bother them to write on paper. So, this will really help us […] We are remaining […] with other five sub counties, whose CHVs have not been digitised.*

Examples like the one above, which utilize mobile health (mHealth) technologies to support data capturing, were implemented in some sections of the county. This suggests that such interventions were potentially in their pilot phase or were donor-driven, raising questions about their sustainability.

**Human resource constraints to monitor indicators.** A recurring theme across counties was that the quality of DHIS2 data was undermined by a shortage of human resources and limited capacity among available staff to monitor indicators.

In County 10, participants noted that a shortage of HRIOs resulted in healthcare providers without adequate training in data capturing, performing multiple roles including maintaining records and report writing. In County 8, participants pointed out a substantial backlog of unprocessed data due to staff shortages. Across counties, participants thus echoed that increasing the number of available HRIOS to capture data at all levels of service delivery is crucial to enhance the quality of data uploaded into the DHIS2 system.

*in terms of data quality, we need at least to have […] clerical officer or HI, or, or […] health information officers at most of our levels of service deliveries, […] so that we can, we can improve the quality of data that […] that […] we are uploading in our system. (Deputy Director of Health, County 10)*

Further, in County 10, participants highlighted the limited capacity of HRIOs to utilise reporting tools as a result of not receiving training on some aspects of data management. Participants in Counties 10, 13 and 15 underscored the need to strengthen the capacity of both health professionals, including HRIOs in data handling at the health facility level to improve documentation and monitoring processes.

*There are so many reporting tools that are brought from the national or wherever, but people are not sensitised [on] what to record [on] those -- those tools. And finally, we'll have uh…uh… data that is not the correct one, because people might be thinking what to record. So, because I think when we have uh…new tools, we need a sensitization, so that people can know exactly what am I recording, what will I be reporting. (Primary Health Coordinator, County 10)*

Overall, participants highlighted the need to strengthen stakeholders' capabilities, through recruitment of additional HRIOs as well as the provision of regular training, including of HCPs' on the use of data capturing tools to improve DHIS2 data accuracy.

## Discussion

While application of information technology is crucial for strengthening HIS, the effective use of data for decision making requires the entire socio-technical system to function cohesively. This study's findings highlight socio-cultural elements such as the practice of reviewing DHIS2 data - primarily at higher levels of health management, supported evidence-based decision making. However, failure to follow up actions from data review meetings meant that data did not always translate to actions. Moreover, a lack of access to the DHIS2 system and information by some stakeholders, e.g., actors at the facility level and political leaders, contributed to a weak sense of data ownership and engagement with it. Technical factors such as DHIS2 data synthesis facilitated advocacy efforts. However, this potential was constrained by limited technical capacity of CHMTs to perform data analysis. Finally, environmental/contextual factors shaped stakeholders' perceived inaccuracy of DHIS2 data, reflecting the level trust among stakeholder in health data. This can arguably influence their willingness to use such data for decision making. Perceived inaccuracy of DHIS2 data was seen to be due to inconsistencies in reporting tools used across counties following the devolution of health governance, non-representativeness of DHIS2 data due to inadequate mechanisms to trace patient transfers and account for service provision by private providers, and human resource shortages and capacity constraints to capture data.

In this section, we discuss the implications of these findings for closing the feedback loop within a socio-technical system to enhance the effective use of DHIS2 data for decision making and advocacy in Kenya's decentralized health system.

The need to strengthen practices that support data ownership and engagement emerged as a key priority. CHMT members' experiences revealed practices such as review of DHIS2 data that can support evidence-based decision making. Yet, these practices were undermined by long reporting intervals and lack of follow-up actions. This highlights the need for stronger accountability measures to ensure that data is not only reviewed but translates into concrete actions, coupled with on-site mentorship and support on different approaches to data use [35,36]. Findings also suggest a need to entrench a sense of data ownership among various stakeholders in health. Similar to experiences from other SSA contexts [11], this study found that DHIS2 generated information was used to guide resource allocation, planning, and advocacy. A significant gap that has been highlighted in this, and other studies is that there is limited data use at the facility level [13,15,37]. Participants cited lack of access to the DHIS2 system by actors at the facility level as a contributing factor. In Kenya, previous studies have linked perceptions of limited data ownership and use at the facility level to hierarchical arrangements favouring ministry-level senior officers' access to DHIS2 platform [37]. Indeed, in the 22 health facilities in Kenya that were examined by Kihuba et al., [17] only 19% of senior managers in hospitals had access to the DHIS2 platform. While participants in this study did not address the reasons that underlie limited access to DHIS2 at the facility level, our

experience implementing the CEBA project and working in the study context suggests that a lack of credentials to access DHIS2, limited awareness about the system, lack of intrinsic motivation and capacity to access the system may be contributing factors. More research is needed to systematically uncover these factors. Findings suggest the need for mapping of stakeholders who need access to the DHIS2 system, defining appropriate levels of access, and identifying barriers to access and addressing them.

Evidence from various SSA contexts suggests the potential to expand engagement with data among a wide range of stakeholders in health decision making by translating data into outputs that are easy to interpret [13–15]. This is especially so for political leaders who, in this study, did not engage in depth on their use of data for decision making. The few that did cited barriers to access and reliance on health departments for information. Notwithstanding, narratives by CHMT members who have a clear mandate to perform data analysis [31] suggested limited capacity to synthesize DHIS2 data. Our findings resonate with quantitative evidence suggesting underutilisation of DHIS2 analytical tools [13,15–17]. Yet, a recent systematic review of interventions to address challenges in routine HIS, such as DHIS2 in LMICs, identified data analysis and display among the least addressed processes [38]. Both our findings and previous literature highlight a clear need for increased investment in analytical capacities of CHMT members [22]. Participants in our study also noted the need to build research capacity at the county level by establishing research departments to provide research-specific technical support such as data analysis, synthesis, visualisation and interpretation to health workers.

Addressing gaps in data collection processes, focusing on strengthening coordination of public and private service providers and linkages with CHWs is key to improving DHIS2 data accuracy. Participants highlighted gaps in DHIS2 data relating to underreporting/overreporting of community health data. In both Malawi and Kenya, Regeru et al., [39] found that only 15% of data reported by CHWs matched consistently with reports by their supervisors. This study sheds light on some of the dynamics that underpin such discrepancies, including the migration of service users between health facilities and inadequate mechanisms to document these movements, and exclusion of community incidents, e.g., of malnutrition in contexts where health facilities are underutilised. Another significant source of data gaps identified by participants in this study was the failure to integrate private service providers, such as private pharmacies, into health data reporting systems. In Kenya, private facilities account for nearly 50% of health facilities [40]. Private pharmacies play a crucial role in addressing the unmet need for contraception in Kenya and other SSA contexts. Commercial drug sellers including pharmacies provide contraceptives to at least one in five young people aged 15–24 across 33 SSA countries [41,42]. Our findings suggest that these contributions are inadequately captured, if any, in DHIS2 data. These gaps call for measures by national and county governments to link data collection systems of private and public health service providers and to strengthen data quality audits with CHWs [20,39].

Our findings shed light on the broader context influencing DHIS2 data accuracy and trust in it among stakeholders in health. In the 15 Counties that were included in this study, participants described the impact of devolution of health governance on resources needed to support HIS. Specifically, occasional shortages of reporting registers, led to the use of outdated registers or improvisation of registers. Unavailability of reporting tools across health facilities in Kenya has been found to be due to confusion over responsibilities, such as whether printing of reporting tools should be handled by the national or county governments, and widespread stockouts [7]. This results in some counties continuing to use old registers or resorting to improvisation which impact negatively data accuracy [43]. These experiences highlight supply chain management challenges relating to data collection tools that need to be streamlined at both county and the national government levels [43].

HRIOS are responsible for capturing and entering data to DHIS2 [31,32]. Kihuba et al., [17] established that only 47% of recommended records officer positions were filled in the 22 facilities they examined in Kenya. Despite these findings, they found that found HIS departments at the facility level to be grossly underfunded. Participants in this study indicated that dependence on a limited pool of HRIOs necessitated task-shifting to other HCPs with limited training in data management and familiarity with DHIS2. They found this to affect DHIS2 data accuracy. Findings show the need for greater

investment in HIS at the County level, specifically allocation of resources by both county and national governments towards recruitment of a sufficient number of HRIOs.

**Strengths and limitations**

This study discusses dynamics that shape perceptions of accuracy of DHIS2 data and its use for decision making. Grounding the interpretation of findings in a sociotechnical lens offers a more critical understanding of these dynamics.

This study has several limitations. Firstly, variation in participation across counties was influenced by availability and willingness of key informants to participate in the study during the data collection period. Across counties, CHMT members were more willing to participate, compared to other targeted participant groups. Recruiting legislators specifically proved challenging. Even though they were assured that their responses would remain confidential, and that the purpose of the study was to gain insights rather than expose shortcomings, some declined to participate while others requested to review interview questions prior to the interview. Few legislators who participated engaged in depth in the discussion on data use. Their contributions, although not detailed- offered useful insights from actors whose experiences on DHIS2 data use are underexplored in existing literature. Future studies can explore more appropriate ways to include political leaders. Their perspectives can facilitate understanding of capacities needed by different stakeholders in health to use data for decision making.

**Conclusions**

We explored the perceptions of DHIS2 data quality and reliance for decision-making among PHC/RMNCAH+N stakeholders across 15 counties in Kenya, highlighting several sociocultural, technical and contextual dynamics that impact data utility. Our findings indicate that while there is a strong practice of reviewing DHIS2 data at higher levels of health management. Limited access to information by political leaders and to the DHIS2 system particularly at the health facility level, limitations in analytical capacity of health management teams and failure to follow up actions after review meetings hindered data-driven making. Trust in DHIS2 data which can impact willingness to rely on it to make decisions was undermined by perceived inaccuracies. This was because of use of inconsistent tools for data reporting across counties, limited integration of private service providers into the DHIS2 system, and human resource constraints to monitor indicators.

Addressing these challenges through investment in analytical skills of health management teams and health, expanding data access to DHIS2 among all relevant stakeholders, fostering a culture of data ownership and use at the facility level and among political leaders, and strengthening accountability measures are critical to enhance data-driven decision-making.

Targeted interventions to improve data accuracy are necessary, including establishing stronger coordination mechanisms between public, private, and community health actors, and enhancing data collection processes to accurately capture service delivery. Additionally, the study underscores the need for greater investment in health information systems at the county level to support routine health data collection and reporting.

Overall, this study contributes to the growing body of evidence on DHIS2 data use in sub-Saharan Africa and provides practical recommendations for improving PHC/RMNCAH+N data quality and utilisation in Kenya. Addressing the identified gaps can enable more robust use of DHIS2 data, ultimately enhancing resource allocation, planning, and service delivery for better health outcomes.

**Supporting information**

**S1 Fig. County-level performance on selected RMNCAH+N tracer indicators for the 15 counties included in the study over four quarters, between quarter one and four of 2022.** Each county's performance on PHC/RMNCAH+N indicators is presented on a separate page, organised in alphabetical order. The county name is displayed at the top right

of each page, and the geographic location of the county within Kenya is indicated on the top left using a small map with the county shaded in yellow. Each county's performance on selected PHC/RMNCAH+N indicators (captured as a percentage on the Y axis) across the four quarters (on the x -axis) is presented using a set of graphs. The following indicators are presented. • Fully immunized child: Proportion of children under one year that are fully immunized. • ANC4: Proportion of pregnant women with antenatal clinic attendance of at least four visits. • Vitamin A supplementation: Proportion of children under 5 years that have received vitamin A supplements. • Skilled birth attendance: Proportion of deliveries attended to by skilled health personnel • HPV vaccine uptake: Proportion of girls aged 10–14 years that have received the HPV vaccine. • Postnatal care: Proportion of women receiving postnatal care within 6 weeks after delivery. • Family planning uptake: Proportion of women of reproductive age 15–49 receiving modern family planning services.
(PDF)

## Acknowledgments

We express our gratitude to the Ministry of Health and participating County Governments, for granting permission to conduct this study. We also acknowledge the contributions of the research assistants who supported data collection. Finally, we sincerely appreciate the study participants for their invaluable cooperation and insights throughout the study.

## Author contributions

**Conceptualization:** Conrad Wanyama, Dan Odallo, Nelly Bosire, Michuki Maina, Irene Amadi, Ambrose Agweyu, Fred Were.

**Data curation:** PHOENE Mesa Oware, Gregory Omondi, Celestine Adipo.

**Formal analysis:** PHOENE Mesa Oware, Celestine Adipo, Mohamed Adow, Conrad Wanyama.

**Investigation:** Celestine Adipo, Ambrose Agweyu, Fred Were.

**Methodology:** PHOENE Mesa Oware, Ambrose Agweyu, Fred Were.

**Project administration:** Gregory Omondi, Irene Amadi, Cynthia Amisi, Fred Were.

**Supervision:** PHOENE Mesa Oware, Gregory Omondi, Dan Odallo, Ambrose Agweyu.

**Visualization:** PHOENE Mesa Oware.

**Writing – original draft:** PHOENE Mesa Oware, Ambrose Agweyu.

**Writing – review & editing:** Conrad Wanyama, Dan Odallo, Nelly Bosire, Lawrence Okong'o, Michuki Maina, Jalemba Aluvaala, Peter Ngwatu, David Githanga, Doris Kinuthia, Irene Amadi, Grace Rukwaro, Linet Kerubo, Cynthia Amisi, Allan Govoga, Janette Karimi, Ali Kahtra, Andrew Mulwa, Patrick Amoth, Ambrose Agweyu, Fred Were.

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
