## [Decision Letter · Decision Letter 0]

2 May 2025

PGPH-D-25-00724

Quality and Utilisation of DHIS2 data for health decision making and advocacy in Kenya: A Qualitative Study

Dear Dr. Oware,

Thank you for submitting your manuscript to PLOS Global Public Health. After careful consideration, we feel that it has merit but does not fully meet PLOS Global Public Health’s publication criteria as it currently stands. Therefore, we invite you to submit a revised version of the manuscript that addresses the points raised during the review process.

We look forward to receiving your revised manuscript.

Kind regards,

Hannah Tappis, DrPH, MPH

Academic Editor

Journal Requirements:

1. We note that you have indicated that there are restrictions to data sharing for this study. PLOS only allows data to be available upon request if there are legal or ethical restrictions on sharing data publicly. For more information on unacceptable data access restrictions, please see http://journals.plos.org/plosone/s/data-availability#loc-unacceptable-data-access-restrictions.   Before we proceed with your manuscript, please address the following prompts:  a) If there are ethical or legal restrictions on sharing a de-identified data set, please explain them in detail (e.g., data contain potentially identifying or sensitive patient information, data are owned by a third-party organization, etc.) and who has imposed them (e.g., a Research Ethics Committee or Institutional Review Board, etc.). Please also provide contact information for a data access committee, ethics committee, or other institutional body to which data requests may be sent.  b) If there are no restrictions, please upload the minimal anonymized data set necessary to replicate your study findings to a stable, public repository and provide us with the relevant URLs, DOIs, or accession numbers. For a list of recommended repositories, please see https://journals.plos.org/plosone/s/recommended-repositories. You also have the option of uploading the data as Supporting Information files, but we would recommend depositing data directly to a data repository if possible.  We will update your Data Availability statement on your behalf to reflect the information you provide. 2. Please provide separate figure files in .tif or .eps format. For more information about figure files please see our guidelines:  https://journals.plos.org/globalpublichealth/s/figures https://journals.plos.org/globalpublichealth/s/figures#loc-file-requirements

Additional Editor Comments (if provided):

Reviewers' comments:

Reviewer's Responses to Questions

**Comments to the Author**

1. Does this manuscript meet PLOS Global Public Health’s publication criteria ? Is the manuscript technically sound, and do the data support the conclusions? The manuscript must describe methodologically and ethically rigorous research with conclusions that are appropriately drawn based on the data presented.

Reviewer #1: Yes

Reviewer #2: Yes

Reviewer #3: Yes

2. Has the statistical analysis been performed appropriately and rigorously?

Reviewer #1: Yes

Reviewer #2: Yes

Reviewer #3: N/A

3. Have the authors made all data underlying the findings in their manuscript fully available (please refer to the Data Availability Statement at the start of the manuscript PDF file)?

Reviewer #1: No

Reviewer #2: Yes

Reviewer #3: No

4. Is the manuscript presented in an intelligible fashion and written in standard English?

Reviewer #1: Yes

Reviewer #2: Yes

Reviewer #3: Yes

5. Review Comments to the Author

Reviewer #1: Thank you so much for this well-written article; I don’t have many comments. May you consider the following

1. Remember to include full words before any acronym is used and that this is consistent throughout the document

2. In Line 177 and Line 216, the sentence starts with a number, write in words like “four” or “fifty-four”. This applies anywhere in the document. Again, if the number is less than 10, it should appear in words

3. In the methods section, include a “reflexivity statement” or paragraph as this is qualitative research

4. I may have missed it but how many people declined to participate in the study? Could you elaborate?

5. Did you pilot the interview guide, a sentence or paragraph would be helpful here to ensure the validity of the tool used

6. What was the duration of the interviews?

7. Include the statement on data availability despite being qualitative research

Reviewer #2: Abstract: The abstract is clear and easily comprehensible. No major revisions are required.

Introduction: The introduction is well-structured and clearly written. No revisions are necessary.

Methodology:

Study Site and Participants

• Line 153: Replace “purposely” with “purposively” to reflect the appropriate sampling terminology.

• Line 153: There is an inconsistency in the number of Key Informant Interviews (KIIs). The abstract states 88 KIIs, while the methodology section mentions 89. Please reconcile this discrepancy and ensure consistency throughout the manuscript.

Coding and Analysis Approach

• Lines 180 and 187: There appears to be a methodological inconsistency regarding the coding approach. If inductive coding was employed, the presence of an initial codebook suggests a deductive framework. To improve clarity and methodological rigor, consider restructuring this section as follows:

"An initial codebook was developed based on the research objectives and preliminary transcript review. Researchers were trained using this initial framework, and inter-coder reliability was assessed to ensure consistency. Subsequently, inductive coding was applied to allow for the emergence of new themes. These emerging codes were reviewed and validated by senior researchers to enhance credibility."

Results:

• Lines 216–259: The descriptive profile of the study participants is currently embedded in the narrative. For clarity and conciseness, consider presenting this information in tabular form (e.g., an upgraded Table 1). Additionally, participant profiles are more appropriate within the Methodology section than the Results.

• Line 230: Correct to “3 County Nursing Officers” to reflect proper pluralization.

• Table 2: There is redundancy between the organizing theme “1a” and its corresponding basic theme, which currently appear identical. This should be revised for clarity. The ‘Basic Theme’ column should briefly summarize the specific topics or sub-themes covered under each organizing theme. Furthermore, this table, like Table 1, should be presented in the Methodology section to align with standard scientific structure.

Discussion: The discussion is well-articulated and effectively synthesizes the findings. No changes required.

Reviewer #3: Quality and Utilisation of DHIS2 data for health decision making and advocacy in Kenya: A Qualitative Study

Review report

Title

-Perhaps it should read as “perceived quality and ....”

Introduction

-The authors highlight valuable insights about the need for strengthening health information systems to support progress tracking. Authors also highlight that DHIS2 improvement is one of the efforts toward addressing health system challenges to having robust health information systems, and go ahead to provide a historical perspective of DHIS2 in Kenya.

-However, given the growing evidence on routine health information systems, including DHIS2, the authors do not provide this context to their work to highlight what is already known about experiences and factors influencing use. Such information would help situate the present work better.

Methods

-Could the authors clarify/confirm the type of counties included? A few characterising elements within the main text – e.g., urbanicity and population size range?

-Regarding the study participants, I am curious to know if the identification of the participant groups by the project/authors was informed by some form of stakeholder mapping, to ensure key actors in the DHIS2 were represented, and in line with key components of the data system for DHIS2? Also, it would be helpful to add a sentence for each participant category highlighting their role concerning data for better understanding who could be left out. For example, it's not clear (from the introduction or methods) who collects and/reports the DHIS2 data and whether these were part of the 89 key informants.

-Regarding data collection, the point that participants were shown graphs showing performance ... is it possible that this could have influenced the responses, depending on what they wanted to be true about their county? Desirability?

-It is not clear how the thematic network analysis was applied.

Results

-It is good that the counties were coded. However, I wonder if these counties would have more than one director or CEO of a hospital. This specification, especially in Table 1, might be implicating in a way. Perhaps the authors should consider revising this table. The text provides better anonymity; maybe the summary table should draw from this text?

-Pg 10, L248 – there is a typo, the ‘w’ should not be there?

-I note that the abbreviation CHMT is not defined at first use which seems to be on pg 8, L194? Authors should confirm this and address it if it is the case.

-Consider providing some more (background) information on the flow of DHIS2, including frequency of reporting. Are all data elements reported monthly? Does the process differ across counties or its fundamentally the same as guided by the national ministry of health.

-Under capacity to synthesize... authors highlight the issue of reliance on HRIOs but what is the relationship between these officers and the CHMT? same comment applies to the other participants or stakeholders mentioned in participant responses. goes back to the comment to provide additional information on the structure of the DHIS2 in Kenya or the counties and the connections between different elements and actors.

-Pg20, L425 implies a focus on the dimension of “accuracy” regarding ‘data quality’. Could the authors clarify how ‘accuracy’ was conceptualized in this study?

-The results in this sub-theme seems to highlight factors potentially affecting data quality, but the specific perceptions of accuracy are not explicit. How did they determine the accuracy of the data, did they perceive the data to be accurate, or not, or conditional?

-The sub-theme ‘devolution’ – authors should consider revising this subheading, perhaps make it more specific in light of the narrative below it.

-Same comment for inadequate data collection mechanisms. – the issues seems to be about representativeness of the DHIS2 data rather than mechanisms. Authors should consider revising this.

Discussion

-Pg26, L567-568: “differentiated access to reporting tools following devolution ...” in results authors use different phrasing “... county had not yet developed its own data capturing registers following devolution...’ (Pg21, L435-436). These two statements do not reflect the same thing and authors should consider harmonising the language around this. “access to” implies someone else is responsible for making the tools available, while the latter statement implies the counties are responsible for making the tools are available. Which is which?

-Pg26, L571: authors use this phrase “data gaps” but in the results it was not very clear what these gaps were. Perhaps authors give a summary of the key gaps they are referring to somewhere in the results. Similarly, I wonder how the authors came up with the “perceived unreliability of DHIS2 data...” statement in the same sentence. This was not highlighted in the results, the actual perceptions of the accuracy?

-From the perspective of key issues influencing use of DHIS2 data by ‘county-level’ stakeholders, the point highlighted on Pg27, L583-584 seems a bit distal and inadequately related to the behaviour of the actual participants in the present study. Is it that because they think that facilities are not using DHIS2 data, they also do or do not use it? Yes data ownership at facility level could be an issue, though this was not explored in this study, but to what extent or in what ways do the actual participants own the data? Authors should look into this argument.

-In the same paragraph, authors point to expanding access to DHIS2 to more stakeholders (as suggested by their participants). Can the authors reflect on what the implications of such an action could be? Data protection, quality control, integrity, etc.? How do they recommend this can be done? What alternatives could be considered?

-The 3rd paragraph of the discussion, the one beginning on pg27, L593, authors should also consider revising it. The authors state that county-level legislators had limited input, which might also apply to other non-CHMT participants. In deed responses of these participants are scanty in the results. It would be good to know what they had to say, however little. Is it that they lacked knowledge about DHIS2, perceived no need for DHIS2 in their lines of work, had never interfaced with DHIS2 data in their line of work? Authors should reflect more on the statement made on pg27, L596 starting with “limited ability of.....” it would be good if authors make the argument of this whole paragraph from the perspective of these non-CHMT participants. What needs did they express with regard to DHIS2? What is relevant for them? Etc. Otherwise, the mentioned engagement platforms that were mentioned by CHMT members might not be applicable.

-The presentation of the issue on devolution should be revised. Shortage of registers or outdated registers can happen across different contexts. What seemed to be an issue was that the counties hadn’t developed their registers, pointing to issues of (administrative) readiness or capacity for the counties and probably what support counties had to function effectively under the devolved system. It is interesting that the availability of tools is an issue over a decade after the devolved system was enforced. But as previously highlighted, authors should clarify the role of counties in the tools/registers, then improve this argument.

-The point about task-shifting: which particular healthcare providers are authors implying that should be doing similar work to the HRIOs? What is the role of HRIOs and HCPs with regard to DHIS2 data? Which tasks exactly should be shifted? Why have health systems recruited HRIOs? The study quoted by the authors in this regard points to the inadequate numbers potentially due to limited recruitment secondary to financial constraints, not a lack of? Could the authors reflect on this issue more and revise accordingly. If its really task-shifting (from the HRIOs), should be clear to who, what exactly and possible implications of this on data quality and subsequent use.

-Pg31, L673: the authors could enhance their integration of the sociotechnical lens in the the interpretation of their work. The current discussion does not reflect it much.

-Pg31, L677: the word should be ‘counties’?

-Could the authors comment on excluding (other) facility managers, HRIOs, and those who produce (collect and report) DHIS2 data?

Conclusion

-If some of the above changes are effected, revising this section might be necessary, for example this statement “persistent data quality gaps and limitations in analytical capacity hinder its effective use, particularly at the health facility level. These challenges are compounded by differentiated access to data reporting tools post-devolution, limited integration of private service providers, and human resource constraints”.

-The first two paragraphs should be integrated to reduce unnecessary repetition.

-Regarding the interventions – authors should be clear on who should effect the recommended strategies.

-What could future research do in light of the study's findings?

Other general comments

-At what point do the county-level decision makers interface with the DHIS2 system or data cycle

-It is likely that the different participant groups have different needs regarding utility of DHIS2 data. It would have been good to highlight differences in perceptions (where possible) across the different groups – technical people, political persons, regulation and the non-state actors. Is it that the authors did not identify anything in this regard.

-The authors highlight two aspects – use of DHIS2, which was primarily linked to capacity issues. And then perceived factors influencing data quality. However, these two elements have not been well linked to provide a more comprehensive picture. In other words in what ways did their perceptions regarding quality (beyond the factors possibly affecting quality) influence their use of the data?

6. PLOS authors have the option to publish the peer review history of their article (what does this mean? ). If published, this will include your full peer review and any attached files.

**Do you want your identity to be public for this peer review?** For information about this choice, including consent withdrawal, please see our Privacy Policy .

Reviewer #1: **Yes: ** Moses Banda Aron

Reviewer #2: No

Reviewer #3: No

---

## [Editor Report · Decision Letter 1]

23 Jul 2025

Perceived Accuracy and Utilisation of DHIS2 data for health decision making and advocacy in Kenya: A Qualitative Study

PGPH-D-25-00724R1

Dear Dr Oware,

We are pleased to inform you that your manuscript 'Perceived Accuracy and Utilisation of DHIS2 data for health decision making and advocacy in Kenya: A Qualitative Study' has been provisionally accepted for publication in PLOS Global Public Health.

Best regards,

Hannah Tappis, DrPH, MPH

Academic Editor